# Finite-Time Thermodynamic Modeling and Analysis of Seawater Acidification Process in Electrochemical Acidification Cell

**DOI:** 10.3390/e24091192

**Published:** 2022-08-26

**Authors:** Chao Wang, Shaojun Xia, Tianchao Xie

**Affiliations:** College of Power Engineering, Naval University of Engineering, Wuhan 430033, China

**Keywords:** finite-time thermodynamics, electrochemical acidification cell, seawater, hydrogen ion concentration, cell voltage

## Abstract

The unsteady process of the acidification of seawater by using an electrochemical acidification cell (EAC) is studied in this paper. The model of the concentration of hydrogen ions ($H^{+}$) in the effluent seawater and the cell voltage of EAC varying with time and working current are built by applying the theory of finite-time thermodynamics, respectively. The semi-empirical formulas of the concentration of $H^{+}$ in the effluent seawater and the cell voltage under the constant current of the Ionpure EAC are obtained, respectively, by fitting the experimental data of the Ionpure EAC. Then, the simulated data are compared with the experimental data. The total work consumption and average power consumption of the Ionpure EAC are obtained from the semi-empirical formulas. The results show that the semi-empirical formulas can simulate the operation process of the Ionpure EAC well. The validity of the models is verified. The increase of the working current will increase the total work consumption and average power consumption of the Ionpure EAC. The proper current can be selected in engineering practice to achieve different goals, such as high efficiency or low energy consumption. The obtained results can provide some guidelines for the optimal design and optimization of EAC.

## 1. Introduction

A sea-based fuel synthesis process [1,2,3,4,5,6,7,8] refers to the new technology of the catalytical synthesizing of fuel by the carbon and hydrogen captured from seawater by using an electrochemical method. The engineering application of the technology will guarantee the safety of military energy on the ocean, and will also expand the sources of renewable energy. Taking into account the ecological requirements, the sea-based fuel synthesis technology is helpful for sustainable development [9,10] by extracting CO_2_ in seawater. The acidification process of seawater by using an electrochemical acidification cell (EAC) is the fundamental component of the sea-based fuel synthesis process by which the pH value of seawater is lowered to below 6.0; the carbon in the seawater exists in the form of carbonic acid. This is the prerequisite of the extraction of carbon dioxide from seawater. DiMascio et al. [11] and Willauer et al. [12] investigated the initial feasibility of the Nalco and Ionpure EAC by using four kinds of synthetic seawater, and actual seawater from Key West (KW seawater); the results show that the EAC can lower the pH value of seawater to below 6.0. Willauer et al. [13] tested the influence of the operation time and flow rate of seawater on the performance of Nalco EAC experimentally; the results show that the extension of the operation time will increase the internal resistance by 34%. Willauer et al. [14] amplified and integrated the Ionpure EAC, and testified the performance of EAC, e.g., the pH value of the effluent seawater, the operation time, and the recovery rate of CO_2_ and H_2_ in four separated experimentation times. Willauer et al. [15] lowered the ion exchange capacity of the electrode of the EAC and testified the performance; the results show that the inner electrical resistance can be lowered by changing the ratio of ion exchange resin and the inertia material. Willauer et al. [16] used the reverse osmosis water and city water of Key West as electrolytes, respectively, and compared the performance of the EAC; the results show that the city water of Key West can be used as electrolytes and can maintain the performance of the EAC under some operating parameters. Based on Refs. [11,12,13,14,15,16], Wang et al. [17] built the grey models of EAC considering single factors and multi-factors and investigated the relation between the performance of the EAC and the working time and electrical current, voltage, and resistance.

The electrochemical acidification cell is the electrolytic cell which transfers electrical energy to chemical energy. Studies of the electrochemical characteristics of the electrolytic cell and fuel cell can help establish the physical model of the electrochemical acidification cell. Udagawa et al. [18] established the electrochemical model of the intermediate temperature solid oxide electrolysis cell, considering the overpotential, and investigated the influence of overpotential on the performance of the cell. Ni et al. [19,20] established the electrochemical models of the solid oxide steam electrolysis cell and investigated the influences of parameters on the energy and exergy efficiency of the process. Finite-time thermodynamics (FTT) [21,22,23,24,25,26,27,28,29,30] have been applied for modeling and performance optimizations of various thermodynamic, chemical, and economic processes, devices, and cycles since 1975. Sieniutycz et al. [31,32,33,34,35] established the physical model of the unsteady state of the multistage electrochemical systems of fuel cells, and analyzed the influence of irreversible losses on the performance of the cell. Xia et al. [36] investigated the optimal current paths for a class of generalized electrochemical reaction systems. Atlsa et al. [37] established the electrodialysis model to predict spatial variations of salt and water ion concentration in an ED cell. Dykstra et al. [38] investigated the pH changes in electrochemical systems for the water desalination process. Abdalla et al. [39] studied the voltage loss at the cathode, the anode, and across the membranes, respectively, in a desalination fuel cell. Khalla et al. [40] investigated the available energy and electricity production of a desalination fuel cell under two operational modes.

Refs. [11,12,13,14,15,16] studied the process of lowering the pH value of seawater by using an EAC experimentally. Ref. [17] established the grey model of the EAC and simulated the working process, but the mathematical model is not the unified model. The physical and mathematical models should be built to describe the process quantitatively based on the physical and chemical mechanism. Based on Refs. [11,12,13,14,15,16,17], this paper will establish the physical and mathematical models of the concentration of hydrogen ions ($H^{+}$) in the effluent acidic seawater and the cell voltage related to working current and time by applying the theory of FTT; and obtain the semi-empirical formulas of the concentration of $H^{+}$ in the effluent acidic seawater and the relation between cell voltage and working current and time with constant working electrical current by fitting the parameters in the semi-empirical formulas based on the experimental data of the Ionpure electrochemical acidification cell. The simulated data are compared with the experimental data to verify the validify of the model.

## 2. Modeling of the Process

### 2.1. The Concentration of $H^{+}$ in the Effluent Seawater

The schematic of the EAC [11,12,13,14,15,16] is shown in Figure 1. Two cation permeable membranes separate the EAC into three parts: the anode compartment, the central ion exchange (IX) compartment, and the cathode compartment. The deionized water flows into the anode and cathode compartments, and the seawater flows into the IX compartment when the EAC works. Under the effect of DC voltage, the $H^{+}$ close to the cation permeable membrane in the anode compartment passes through the membrane and gets into the IX compartment, and the ${\mathrm{Na}}^{+}$ in the seawater in the IX compartment passes through the membrane close to the cathode compartment and gets into the cathode compartment. The ${\mathrm{Na}}^{+}$ in the seawater is replaced by $H^{+}$, so the seawater is acidified.

The seawater is described as NaCl solution. The main reactions in the EAC are [11,12,13,14,15,16]:

Anode:(1)$$2H_{2}O\to {4H}^{+}+O_{2}+4e^{-}$$

IX:(2)$$4{\left|\mathrm{NaCl}\right|}_{\mathrm{Seawater}}+4H^{+}\to 4{\mathrm{Na}}^{+}+4{\left|\mathrm{HCl}\right|}_{\mathrm{Acidic}\;\mathrm{Seawater}}$$

Cathode:(3)$$4H_{2}O+4{\mathrm{Na}}^{+}+4e^{-}\to 4{\mathrm{NaOH}+2H}_{2}$$

Overall:(4)$$6H_{2}O+4\mathrm{NaCl}\to 4{\left|\mathrm{HCl}\right|}_{\mathrm{IX}}+4\left|\mathrm{NaOH}\right|+2{\left|H_{2}\right|}_{\mathrm{Cathode}}+{\left|O_{2}\right|}_{\mathrm{Anode}}$$

Choosing the IX compartment as the control volume, the changes of $H^{+}$ in the IX compartment are composed of four parts: (1) the inflow of $H^{+}$ in the seawater entering the IX compartment; (2) the outflow of $H^{+}$ in the acidic seawater leaving the IX compartment; (3) the inflow of $H^{+}$ through the membrane from the anode compartment to the IX compartment; (4) the consuming of $H^{+}$ with the ${\mathrm{HCO}}_{3}{}^{-}$ in the IX compartment.

The $H^{+}$ in the seawater entering the IX compartment comes from the water-splitting reaction, and the income of $H^{+}$ from the anode compartment will inhibit the splitting of water in the IX compartment. However, according to the experimental data [14], one can see that the concentration of $H^{+}$ in the outflow seawater is five orders of magnitude larger than in the inflow seawater. This means the change of $H^{+}$ caused by the balance movement of the water-splitting reaction in the IX compartment is very small compared to the amount of $H^{+}$ entering through the membrane. As such, the water-splitting reaction in the IX compartment is ignored to simplify the problem.

Based on the analyses above, the molar balance of $H^{+}$ in the IX compartment is:(5)$$\Delta n_{H^{+},V_{m}}=n_{H^{+},\mathrm{in}}-n_{H^{+},\mathrm{out}}-n_{H^{+},\mathrm{consume}}$$
where $\Delta n_{H^{+},V_{m}}$ is the amount of $H^{+}$ changes in the IX compartment; $n_{H^{+},\mathrm{in}}$ is the amount of $H^{+}$ flowed in from the boundary of IX compartment, which is the sum of (1) and (3) above; $n_{H^{+},\mathrm{out}}$ is the amount of $H^{+}$ flowed out of the IX compartment; and $n_{H^{+},\mathrm{consume}}$ is the amount of $H^{+}$ consumed in the IX compartment. From Equation (5), the molar balance of $H^{+}$ in the IX compartment is:(6)$$V_{m}\eta_{V}\frac{d\bar{c_{H^{+},V_{m}}}}{dt}=\frac{dc_{H^{+},q_{\mathrm{in}}}}{dt}+q_{m}\left(c_{H^{+},\mathrm{in}}-c_{H^{+},\mathrm{out}}\right)-q_{m}c_{T[{\mathrm{CO}}_{2}]}\alpha_{H_{2}{\mathrm{CO}}_{3}}$$

The left part of Equation (6) refers to the change rate of the concentration of $H^{+}$ in the IX compartment, where $V_{m}$ is the volume of the IX compartment, and $\eta_{V}$ is the volumetric efficiency. Since there are inert ceramic particles or strong cation exchange resin in the IX compartment, which occupy some volume, the real volume, $V_{m,\mathrm{real}}$, is smaller than $V_{m}$, so $\eta_{V}(\equiv V_{m,\mathrm{real}}/V_{m})<1$. $\bar{c_{H^{+},V_{m}}}$ is the average concentration of $H^{+}$ in the IX compartment, and $c_{H^{+},\mathrm{in}}<\bar{c_{H^{+},V_{m}}}<c_{H^{+},\mathrm{out}}$. $c_{H^{+},q_{\mathrm{in}}}$ is the amount of $H^{+}$ passing through the membrane into the IX compartment from anode compartment. $c_{H^{+},\mathrm{in}}$ and $c_{H^{+},\mathrm{out}}$ are the concentration of $H^{+}$ in the inflow seawater and outflow acidic seawater, respectively. $q_{m}$ is the volume flow rate of the seawater. $c_{T[{\mathrm{CO}}_{2}]}$ is the total concentration of CO_2_ in the seawater. $\alpha_{H_{2}{\mathrm{CO}}_{3}}$ is the molar fraction of $H_{2}{\mathrm{CO}}_{3}$ in the seawater, which reflects the amount of $H^{+}$ consumed by the ${\mathrm{HCO}}_{3}{}^{-}$. The relation between $\alpha_{H_{2}{\mathrm{CO}}_{3}}$ and $c_{H^{+},\mathrm{out}}$ is [9]:(7)$$\alpha_{H_{2}{\mathrm{CO}}_{3}}={\left(c_{H^{+},\mathrm{out}}\right)}^{2}/\left[{\left(c_{H^{+},\mathrm{out}}\right)}^{2}+c_{H^{+},\mathrm{out}}K_{1}+K_{1}K_{2}\right]$$
where $K_{1}$ and $K_{2}$ are equilibrium constants at 25 °C, $K_{1}=4.47\times 10^{-7}$, and $K_{2}=4.69\times 10^{-11}$.

The calculating errors in the idealized model above come from several aspects: the error of the lumped parameter model, since the concentration of $H^{+}$ in the IX compartment varies complicatedly with time and location; the error of the non-equilibrium ion diffusion process, which is unconsidered in Equation (7). So, $\bar{c_{H^{+},V_{m}}}$ is replaced by $c_{H^{+},\mathrm{out}}$ in the calculation. The proximity coefficient of concentration $\gamma_{c}$ is defined as follows:(8)$$\gamma_{c}=\frac{\bar{c_{H^{+},V_{m}}}}{c_{H^{+},\mathrm{out}}}$$
$\gamma_{c}$ ranges from 0 to 1, and it is assumed to be constant here to simplify the problem.

The amount of $H^{+}$ passing through the membrane from the anode compartment to the IX compartment is a function of time. In the steady working condition, $dc_{H^{+},q_{\mathrm{in}}}/dt$ is proportionate to the working current of the EAC:(9)$$\frac{dc_{H^{+},q_{\mathrm{in}}}}{dt}=60\frac{I}{F}$$
where F is the Faraday constant, and I is the working current of the EAC. In real operation, it takes time for the EAC to reach the steady working condition, and the time varies with the structure, the filling material, and the current of different EAC.

This means the performance of the EAC is influenced by many factors. In this study, we are not interested in the influence of each factor, but instead focus on the change law of the overall performance over time.

The semi-theoretical and semi-empirical equation can be obtained by considering the influence of time on the progress of transmission, $H^{+}$. Equation (9) is modified with the function of time:(10)$$\frac{dc_{H^{+},q_{\mathrm{in}}}}{dt}=60\frac{I}{F}\times f\left(t\right)$$
where t is the working time, $f\left(t\right)\in [0,1)$ in the unsteady working state, and $f\left(t\right)=1$ is the steady working state.

Substituting Equations (7), (8) and (10) into Equation (6), one can obtain the model of the pH of effluent seawater:(11)$$\begin{array}{c} V_{m}\eta_{V}\gamma_{c}\frac{dc_{H^{+},\mathrm{out}}}{dt}=60\frac{I}{F}\times f\left(t\right)+q_{m}\left(c_{H^{+},\mathrm{in}}-c_{H^{+},\mathrm{out}}\right)\\ \;\;\;\;\;\;\;\;\;\;\;\;\;-q_{m}c_{T[{\mathrm{CO}}_{2}]}{\left(c_{H^{+},\mathrm{out}}\right)}^{2}/\left[{\left(c_{H^{+},\mathrm{out}}\right)}^{2}+c_{H^{+},\mathrm{out}}K_{1}+K_{1}K_{2}\right] \end{array}$$

### 2.2. Cell Voltage of the EAC

The cell voltage of the EAC consists of five parts: the theoretical decomposition voltage, the overpotential, the conductor voltage drop, the membrane voltage drop, and the contact voltage drop. The last three voltage drops above can be described by Ohm’s law in order to simplify the calculation, merging them to the equivalent internal resistance, $R_{i}$. The cell voltage can be described as follows [18,19,20,21,22,23,24,25,26]:(12)$$V_{\mathrm{cell}}=V_{\mathrm{de}}+V_{\mathrm{ir}}+V_{\mathrm{loss}}$$
where $V_{\mathrm{cell}}$ is the cell voltage, $V_{\mathrm{de}}$ is the theoretical decomposition voltage, $V_{\mathrm{ir}}$ is the overpotential, and $V_{\mathrm{loss}}=IR_{i}$ reflects the three voltage drops.

The overpotential stems from two parts: the concentration polarization and the electrochemical polarization. The concentration polarization is caused by the mass diffusion, since the applied current added to the electrode produces a concentration gradient near the electrode surface. The concentration polarization in the anode and cathode are [18,19,20,31,32,33,34,35,36]:(13)$$V_{\mathrm{conc}}=\frac{RT}{zF}\ln\left(\frac{i_{d}}{i_{d}-i}\right)$$
where $i_{d}$ is the limiting current density, z is the number of reactive electrons, i=I/A is the current density, and A is the area of the electrode.

The electrochemical polarization is caused by the deviation of the electrode potential from equilibrium, since the applied current causes the unbalance of the current density in the anode and cathode, which destroys the balance of the two electrodes. The electrochemical polarization in the anode and cathode are [18,19,20,31,32,33,34,35,36]:(14)$$V_{\mathrm{ec}}=\frac{RT}{\alpha zF}\ln\left(\frac{i}{i_{0}}\right)$$
where $i_{0}$ is the exchange current density; α is the electron transport coefficient; $\alpha \in \left(0,1\right)$; $\alpha_{a}+\alpha_{c}=1$; and the subscripts, a and c, refer to the anode and cathode.

The concentration polarization and electrochemical polarization exist simultaneously in the real electrochemical process. The overpotential can be described as follows [18,19,20,31,32,33,34,35,36]:(15)$$\begin{array}{c} V_{\mathrm{ir}}=V_{\mathrm{conc},c}+V_{\mathrm{conc},a}+V_{\mathrm{ec},c}+V_{\mathrm{ec},a}\\ \;\;\;\;=\frac{RT}{z_{c}F}\ln\left(\frac{i_{d,c}}{i_{d,c}-i}\right)+\frac{RT}{z_{a}F}\ln\left(\frac{i_{d,a}}{i_{d,a}-i}\right)\\ \;\;\;\;+\frac{RT}{\alpha_{c}z_{c}F}\ln\left(\frac{i}{i_{0,c}}\right)+\frac{RT}{\alpha_{a}z_{a}F}\ln\left(\frac{i}{i_{0,a}}\right) \end{array}$$

Substitute Equation (15) into Equation (12), and the cell voltage is:(16)$$\begin{array}{c} \;\;\;V_{\mathrm{cell}}=V_{\mathrm{de}}+IR_{i}+\frac{RT}{z_{c}F}\ln\left(\frac{i_{d,c}}{i_{d,c}-i}\right)+\frac{RT}{z_{a}F}\ln\left(\frac{i_{d,a}}{i_{d,a}-i}\right)\\ +\frac{RT}{\alpha_{c}z_{c}F}\ln\left(\frac{i}{i_{0,c}}\right)+\frac{RT}{\alpha_{a}z_{a}F}\ln\left(\frac{i}{i_{0,a}}\right) \end{array}$$

The instantaneous power and total work consumption under constant working current are:(17)$$P=IV_{\mathrm{cell}}$$
(18)$$W=\int_{0}^{\tau }Pdt$$

The average power consumption is:(19)$$\bar{P}=\frac{W}{\tau }$$

## 3. Establishment and Verification of Semi-Empirical Formulas

The transfer of $H^{+}$ through the membrane from the anode compartment to the IX compartment is affected by the working current and the operation time, and is also concerned with the structure and filling material of the EAC. Ref. [14] modified a standard, commercially available electrodeionization cell to make it work for the acidification process of seawater. It is nominated as the Ionpure EAC. The seawater flows into the Ionpure EAC at the flow rate, $q_{m}=1.9\;L/\min$, under 30A, 20A, and 10A working current. The pH of effluent seawater and the instantaneous cell voltage were tested in Ref. [14]. The semi-empirical formulas are built based on the experimental data of the Ionpure EAC [14].

### 3.1. The Concentration of $H^{+}$ in the Effluent Seawater

From the experimental data of the Ionpure EAC [14] under 30A, 20A, and 10A current, one can see that the working process of the EAC can be divided into four periods. Under 30A working current, the first period is from 0 to 5 min, and during this period, the pH of effluent seawater raises with time; this is caused by the property of the electrode material. The second period is from 5 to 20 min, and during this period, the pH of effluent seawater decreases with time, and at 20 min, the pH of effluent seawater is below 6. The third period is from 20 to 30 min, and during this period, the pH of effluent seawater decreases sharply to below 3. The fourth period is from 30 to 40 min, and during this period, the pH of effluent seawater decreases a little and approaches the minimum. Under 20A working current, the tendency of the variation of the pH of effluent seawater is similar to the condition of 30A working current, but the time of the second and third period is prolonged to 25 and 35 min, respectively. Under 10A working current, the pH of effluent seawater is lowered to below 6, and during the 40 min, the Ionpure EAC works in the second period. According to Ref. [14], the pH of effluent seawater cannot be lowered to below 3 under 10A working current.

The pH of effluent seawater can be lowered to below 3 by using the Ionpure EAC under 30A working current during the 40 min working period, and then in the 40 to 60 min period, the pH of effluent seawater decreases a little [14]. It can be assumed that the Ionpure EAC can reach a stable working condition at the 40 min basically under 30A working current. The $f\left(t\right)$ in Equation (10) increases monotonically in 5 to 60 min, and the fitting function is chosen as $f\left(t\right)=t^{2}/\left(t^{2}+at+b\right)$. a and b are obtained by substituting the experimental data: a=−44.34, b=2696.69. Figure 2 shows the configuration of $f\left(t\right)$ with time.

Substituting $f\left(t\right)$ into Equation (11), one can obtain:(20)$$\begin{array}{l} V_{m}\eta_{V}\gamma_{c}\frac{dc_{H^{+},\mathrm{out}}}{dt}=\frac{60It^{2}}{F(t^{2}-44.34t+2696.69)}\\ +q_{m}(c_{H^{+},\mathrm{in}}-c_{H^{+},\mathrm{out}})-q_{m}c_{T[{\mathrm{CO}}_{2}]}\alpha_{H_{2}{\mathrm{CO}}_{3}} \end{array}$$
Equation (20) is the semi-empirical model of the concentration of $H^{+}$ in the effluent seawater of the Ionpure EAC.

From the analyses above, one can see that under 20A and 30A working current, the Ionpure EAC can lower the pH of seawater to below 3 in a 40 min working period and reach a stable working condition basically. The semi-empirical formulas of the EAC under 30A and 20A working current are tested below. The function of the boundary value problems of differential equations, “ode15s”, is used in the scientific computing software, Matlab. The beginning of the second period, t=5, when the pH of seawater begins to decrease, is selected as the initial point, and $\eta_{V}\gamma_{c}=0.5$.

The comparisons of simulated data and experimental data under two working currents in 40 min are shown in Table 1. One can see that the relative errors of the simulated data are below 10%, except for 10 min under 30A working current and 15 min under 20A working current. The average relative errors of 30A and 20A working current are 5.06% and 4.98%, respectively.

Figure 3 and Figure 4 show the comparison of the simulated curve and the experimental curve of the pH of effluent seawater in 40 min under 30A and 20A working current, respectively. The full line represents the simulated curve and the dotted line represents the experimental curve. One can see that the trends of the simulated curves under both the two working currents are uniform with the corresponding experimental curves. The simulated curve is below the experimental curve in the second period, and the two curves are very close in the third and fourth periods.

One can see that this semi-empirical formula can simulate the working process of the Ionpure EAC, which can reach a stable working condition in 40 min. This model can help comprehend the working process of the EAC with time and current, and help forecast the pH of effluent seawater in the unsteady working condition. For different EAC with different structures or filling materials, one can obtain the model of the pH of effluent seawater under unsteady working conditions by fitting different $f\left(t\right)$.

### 3.2. Cell Voltage of EAC under Constant Working Current

The limiting current density and exchange current density are relevant to the concentration of the reaction system, the electrode material, the structure of the EAC, and other factors. To study the regulation of the cell voltage of the EAC, the parameters can be selected in proper ranges when there is a lack of experimental data.

From the experimental data of instantaneous cell voltages of the Ionpure EAC [14], one can see that the cell voltage decreases continuously until the pH of effluent seawater is below 3. This is caused by the performance of the strong ion exchange resins. It can be assumed that the decomposition voltage, $V_{\mathrm{de}}$, is basically the same under constant temperature and constant current. The equivalent internal resistance is minimum when the cell voltage is minimum, and remains constant under different currents. Therefore, one can choose the experimental data, $V_{\mathrm{cell},\min}=26.6\;V$ when I=30 A and $V_{\mathrm{cell},\min}=19.4\;V$ when I=20 A, to calculate.

The theoretical decomposition voltage, $V_{\mathrm{de}}$, and the minimum equivalent internal resistance, $R_{i,\min}$, can be obtained by solving the following equations simultaneously:(21)$$\begin{array}{ll} 26.6 & =V_{\mathrm{de}}+30R_{i,\min}+\frac{RT}{z_{c}F}\ln\left(\frac{i_{d,c}}{i_{d,c}-30/A}\right)\\ & +\frac{RT}{z_{a}F}\ln\left(\frac{i_{d,a}}{i_{d,a}-30/A}\right)+\frac{RT}{\alpha_{c}z_{c}F}\ln\left(\frac{30}{Ai_{0,c}}\right)\\ & +\frac{RT}{\alpha_{a}z_{a}F}\ln\left(\frac{30}{Ai_{0,a}}\right) \end{array}$$
(22)$$\begin{array}{ll} 19.4 & =V_{\mathrm{de}}+20R_{i,\min}+\frac{RT}{z_{c}F}\ln\left(\frac{i_{d,c}}{i_{d,c}-20/A}\right)\\ & +\frac{RT}{z_{a}F}\ln\left(\frac{i_{d,a}}{i_{d,a}-20/A}\right)+\frac{RT}{\alpha_{c}z_{c}F}\ln\left(\frac{20}{Ai_{0,c}}\right)\\ & +\frac{RT}{\alpha_{a}z_{a}F}\ln\left(\frac{20}{Ai_{0,a}}\right) \end{array}$$
where $A=497{\;\mathrm{cm}}^{2}$ is the surface area of the electrode, molar constant $R=8.314\;J/\left(\mathrm{mol}\cdot K\right)$, temperature T=298 K, and $z_{c}=z_{a}=4$.

Since the property of the material of the Ionpure EAC is unknown and there are no relevant experiments to test it, one can simplify the situation and study the performance character of the Ionpure EAC qualitatively by assuming that $i_{d,c}=i_{d,a}=i_{d}$, $i_{0,c}=i_{0,a}=i_{0}$, $\alpha_{c}=\alpha_{a}=0.5$. The calculating results of the theoretical decomposition voltage and minimum equivalent internal resistance of the Ionpure EAC under different $i_{d}$ and $i_{0}$ are listed in Table 2.

One can see that $V_{\mathrm{de}}$ decreases and $R_{i,\min}$ increases gradually when $i_{d}$ increases and $i_{0}$ is fixed, and both of them change very little. This indicates that the influences of the value of $i_{d}$ on $V_{\mathrm{de}}$ and $R_{i,\min}$ are small. $R_{i,\min}$ remains constant and $V_{\mathrm{de}}$ decreases when $i_{0}$ increases and $i_{d}$ is fixed. When $i_{0}$ decreases two orders of magnitude, $V_{\mathrm{de}}$ decreases by about 2.5%. One can draw the conclusion that $i_{d}$ and $i_{0}$ only have quantitative influence, and have no qualitative influence on $V_{\mathrm{de}}$ and $R_{i,\min}$. As such, $i_{0}=10^{-6}{\;A/\mathrm{cm}}^{2}$ and $i_{d}=0.15{\;A/\mathrm{cm}}^{2}$ are chosen for the calculation below to investigate the performance of the Ionpure EAC. The minimum cell voltage can be written as:(23)$$\begin{array}{c} \;\;\;\;\;\;\;V_{\mathrm{cell},\min}=V_{\mathrm{de}}+IR_{i,\min}+\frac{RT}{2F}\ln\left(\frac{i_{d}}{i_{d}-i}\right)+\frac{RT}{F}\ln\left(\frac{i}{i_{0}}\right)\\ \;\;\;\;\;\;\;\;=4.75+0.72I+0.013\ln\left(\frac{0.15}{0.15-I/497}\right)\\ +0.026\ln\left(2012I\right) \end{array}$$

The minimum cell voltage is the cell voltage when the performance of the EAC is maximum, which indicates the ion release of strong ion exchange resin is maximum. The relation between the minimum cell voltage and the current is shown in Figure 5. One can see that the minimum cell voltage increases near-linearly with the increase of working current. When the working current increases from 8A to 60A, $V_{\mathrm{cell},\min}$ increases from 10.8V to 48.3V.

Under constant working current, the cell voltage of the EAC decreases first and then increases. One can deem that the variation of the cell voltage is caused by the variation of the equivalent internal resistance with time. The cell voltage under constant working current can be written as:(24)$$\begin{array}{c} \;\;\;\;\;\;\;V_{\mathrm{cell}}\left(t\right)=V_{\mathrm{de}}+IR_{i}\left(t\right)+\frac{RT}{2F}\ln\left(\frac{i_{d}}{i_{d}-i}\right)+\frac{RT}{F}\ln\left(\frac{i}{i_{0}}\right)\\ \;\;\;\;\;\;\;\;=4.75+IR_{i}\left(t\right)+0.013\ln\left(\frac{0.15}{0.15-I/497}\right)\\ +0.026\ln\left(2012I\right) \end{array}$$

The equivalent internal resistances of the Ionpure EAC under 30A, 20A, and 10A currents at 25 °C are calculated and shown in Table 3.

Figure 6 shows the relation between the equivalent internal resistance and the time under different working currents. One can see that the variation of $R_{i}$ can also be divided into some periods: $R_{i}$ decreases first when the EAC begins to work, then increases a little, then decreases continuously to the minimum $R_{i,\min}$, and then begins to increase. This is uniform with the variation period of the pH of effluent seawater: at the beginning, $R_{i}$ decreases and then increases to about 0.85 Ω; it is the flushing process of strong ion exchange resin, and the pH of effluent seawater increases in this period; then, $R_{i}$ decreases to the minimum $R_{i,\min}$, the pH of effluent seawater pH begins to decrease to below 4, and then $R_{i}$ increases and the decrease rate of the pH of effluent seawater decreases. Under 30A and 20A current, $R_{i}$ decreases to the $R_{i,\min}$ in 40 min, and the pH of effluent seawater is lowered to below 4. While under 10A current, $R_{i}$ cannot reach the minimum in 40 min, and the pH of effluent seawater can not be lowered to below 4.

The relation of $R_{i}$ and t is described as:(25)$$R_{i}=k-\frac{t^{2}}{at^{2}+bt+c}$$

The parameters k,a,b,c can be fitted from the experimental data. The results are shown in Table 4.

Figure 7 and Figure 8 show the comparison of the simulated curve and experimental curve of the equivalent internal resistance under 30A and 20A working current, respectively. One can see that except in the first 5 min, the two curves coincide basically. Equation (25) can be used to describe the variation of $R_{i}$ of the Ionpure EAC with time.

In order to study the work consumption of the EAC, the relationship between the equivalent internal resistance and time and working current should be obtained. Constrained by the limited experimental data, the parameters a, b, and c are transformed to the functions of working current, which are available in ranges from 20A to 30A:(26)$$a\left(I\right)=\frac{I-20}{30-20}\times \left(30.6-26.0\right)+26.0=0.46I+16.8$$
(27)$$b\left(I\right)=\frac{I-20}{30-20}\times \left(-1316+1193\right)-1193=-12.3I-947$$
(28)$$\begin{array}{ll} c\left(I\right) & =\frac{I-20}{30-20}\times \left(1.87\times 10^{4}-1.92\times 10^{4}\right)+1.92\times 10^{4}\\ & =-50I+2.02\times 10^{4} \end{array}$$

Substituting Equations (26)–(28) into Equation (25), the variation of $R_{i}$ with time and working current can be obtained as:(29)$$R_{i}\left(I,t\right)=0.85-\frac{t^{2}}{\left(0.46I+16.8\right)t^{2}+\left(-12.3I-947\right)t-50I+2.02\times 10^{4}}$$
The variation of cell voltage with time and working current is:(30)$$\begin{array}{l} V_{\mathrm{cell}}\left(I,t\right)=4.75+0.013\ln\left(\frac{0.15}{0.15-I/497}\right)+0.026\ln\left(2012I\right)+\\ I\left[\;0.85-\frac{t^{2}}{\left(0.46I+16.8\right)t^{2}+\left(-12.3I-947\right)t-50I+2.02\times 10^{4}}\right] \end{array}$$

Figure 9 shows the comparison of the simulated curve and experimental curve of the cell voltage under 30A and 20A working currents. One can see that the simulated curves and experimental curves have some differences in the first 10 min, and then the two curves differ a little. The comparisons of simulated data and experimental data are listed in Table 5. The results show that the average relative errors are 1.14% under 30A working current, and 1.85% under 20A working current. Equation (30) is the semi-empirical formular of variation of the cell voltage of the Ionpure EAC with time and current with working current ranging from 20A to 30A.

According to Ref. [14], the product of working current and time is constant. The actual meaning of it is that the quantity of electricity to lower the pH of seawater to a given value is a constant. The constant is obtained as 900 by the experimental data under 30A working current in Ref. [14]. The working time can be written as:(31)τ=900/I

Substituting Equation (31) into Equations (18) and (19), the total work consumption and average power consumption can be obtained. Figure 10 shows the variation of total work consumption and average power consumption with working current. One can see that both the total work consumption and average power consumption increase near-linearly with working current. When the working current increases from 20A to 30A, the total work consumption raises from 1.14 × 10^6^ J to 1.59 × 10^6^ J, and increases by 39.47%; the average power consumption increases from 423 W to 883W, and increases by 108.74%. One can see that relatively small working current in the proper range can be selected to decrease the energy consumption.

## 4. Conclusions

The unsteady acidification process of seawater by using an EAC is studied in this paper. The model of the concentration of $H^{+}$ in effluent seawater and the cell voltage of the EAC varying with time and working current are built by applying the theory of FTT, respectively. The semi-empirical formulas of the concentration of $H^{+}$ in the effluent seawater and the cell voltage under a constant working current of the Ionpure EAC are established, respectively, by fitting the experimental data in Ref. [14]. Then, the simulated data are compared with the experimental data, and the results show that the difference between the experimental data and the simulated data obtained by the two semi-empirical formulas is relatively large at the beginning of the working process, and then decreases. The average relative errors of the simulated data obtained by the semi-empirical formulas of the concentration of $H^{+}$ in the effluent seawater and the cell voltage of the EAC under a constant working current are below 5% and 2%, respectively. The semi-empirical formulas can simulate the working process of the Ionpure EAC well. The validity of the models is verified. The total work consumption and average power consumption are calculated from the semi-empirical formula of the cell voltage of the EAC. The increase of working current increases the velocity of the transfer of $H^{+}$ from the anode to the IX compartment, so the time to reach a steady working condition is shortened. However, it will increase the total work consumption and average power consumption of the EAC. In engineering practice, proper working current should be selected to achieve different goals, such as high efficiency or low energy consumption. The results obtained can help understanding the working process of EAC and forecasting the working condition of given EAC, meantime, provide theoretical basis for the performance study of EAC.

## Figures and Tables

**Figure 1 entropy-24-01192-f001:**
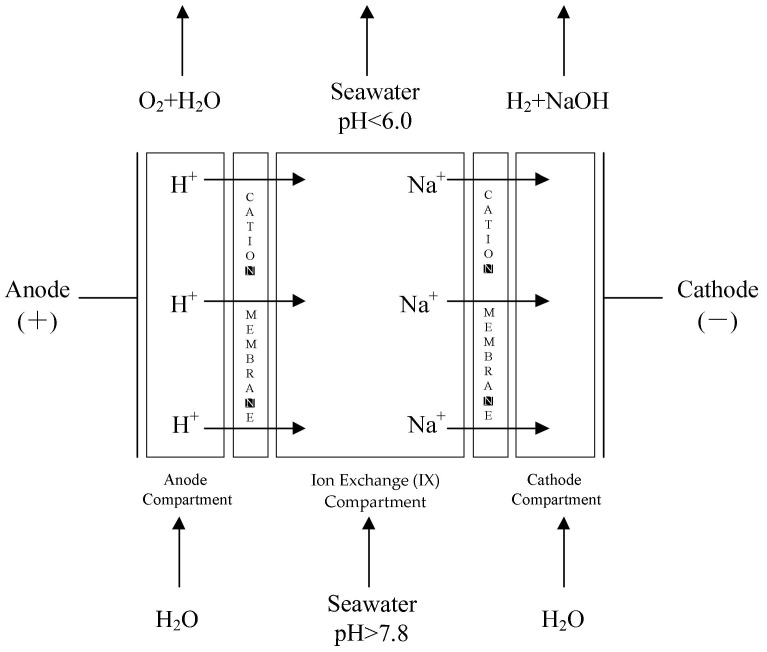
Schematic of EAC.

**Figure 2 entropy-24-01192-f002:**
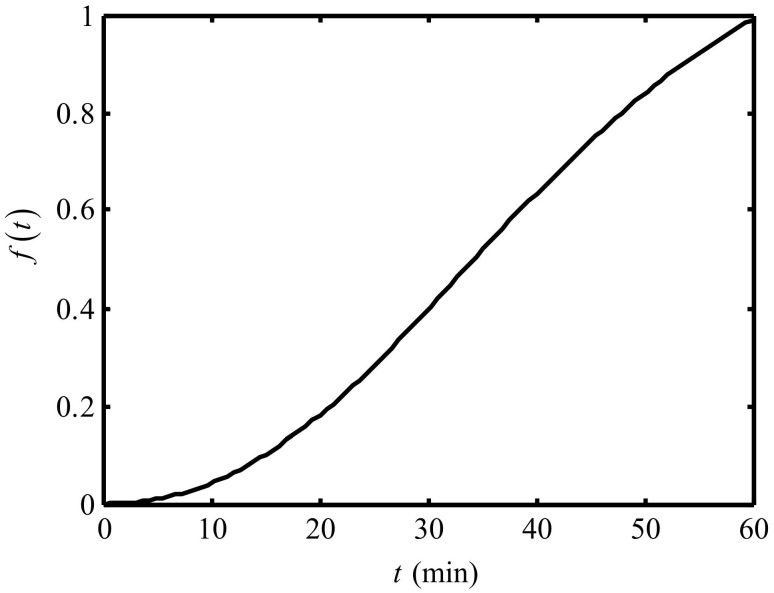
The variation of f(t) versus time.

**Figure 3 entropy-24-01192-f003:**
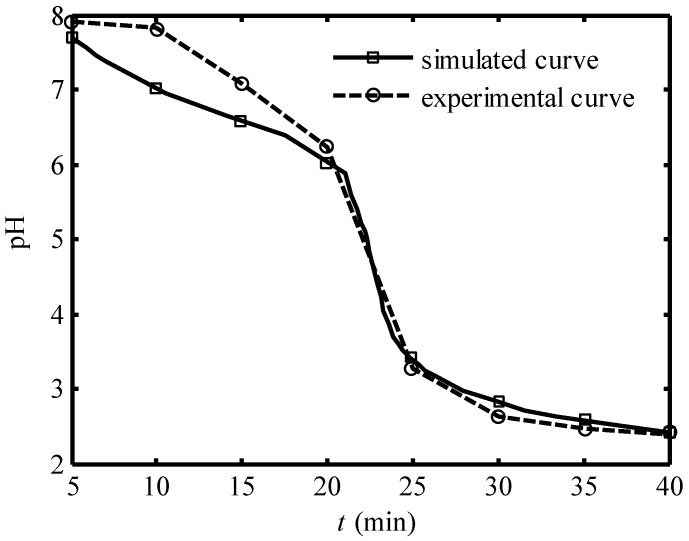
Comparison of the simulated curve and experimental curve [14] of pH of effluent seawater in 40 min under 30A working current.

**Figure 4 entropy-24-01192-f004:**
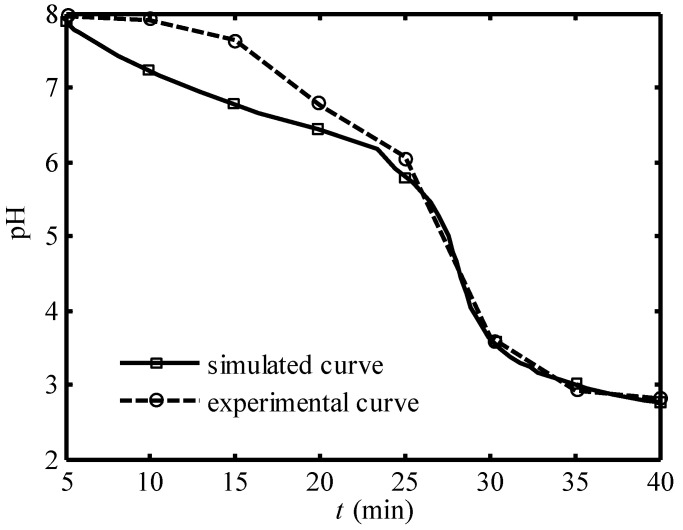
Comparison of the simulated curve and experimental curve [14] of pH of effluent seawater in 40 min under 20A working current.

**Figure 5 entropy-24-01192-f005:**
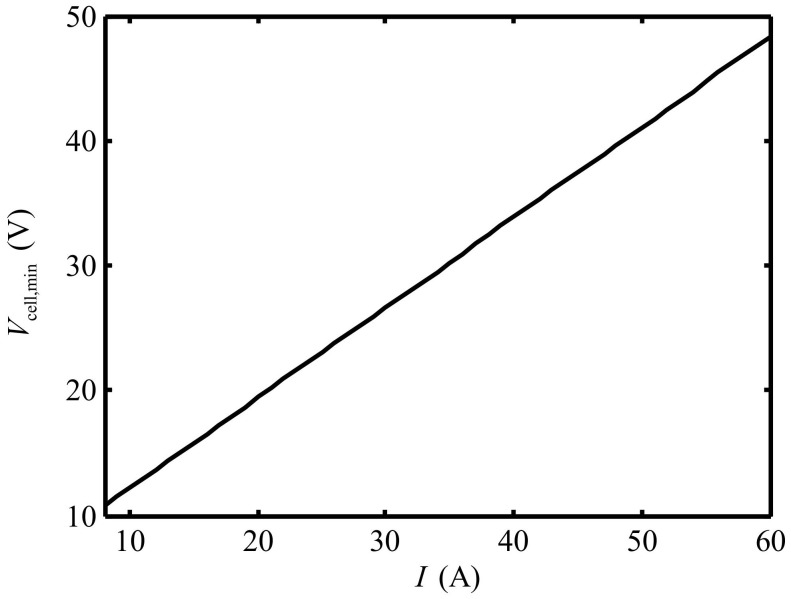
Variation of the minimum cell voltage versus current.

**Figure 6 entropy-24-01192-f006:**
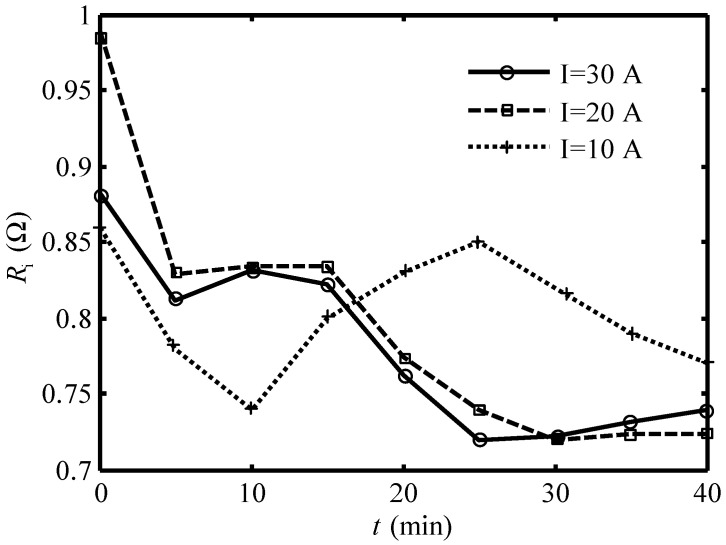
Variation of the equivalent internal resistance versus time under different currents.

**Figure 7 entropy-24-01192-f007:**
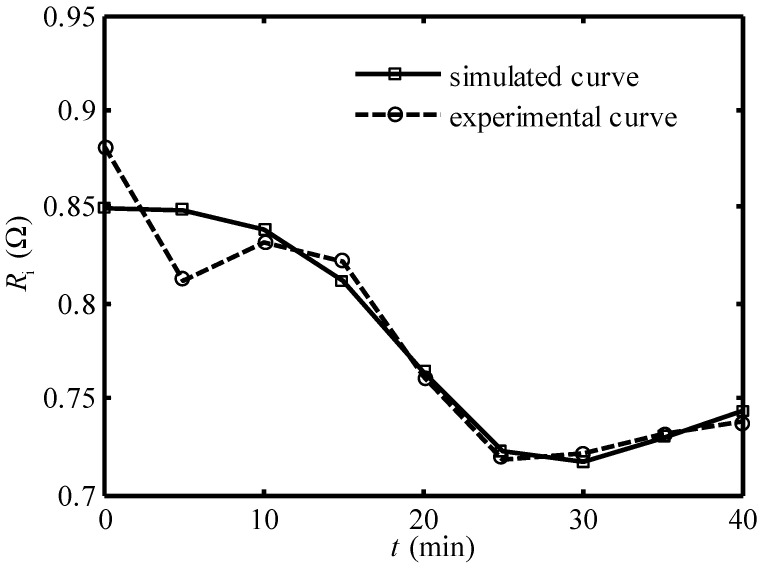
Comparison of the simulated curve and experimental curve [14] of equivalent internal resistance under 30A working current.

**Figure 8 entropy-24-01192-f008:**
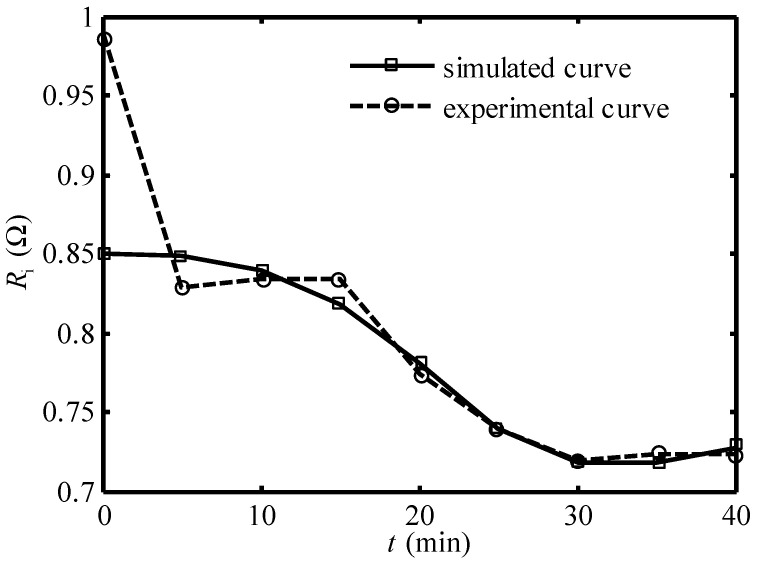
Comparison of the simulated curve and experimental curve [14] of equivalent internal resistance under 20A working current.

**Figure 9 entropy-24-01192-f009:**
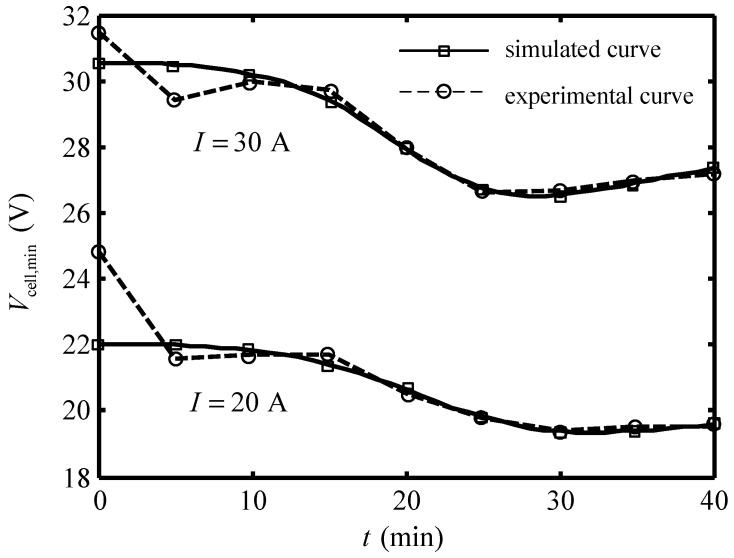
Comparison of the simulated curve and experimental curve [14] of variation of cell voltage versus time.

**Figure 10 entropy-24-01192-f010:**
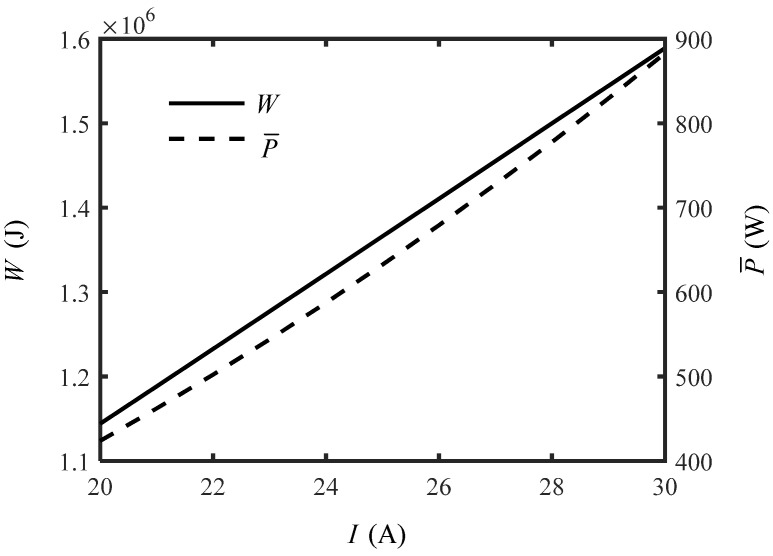
Variation of the total work consumption and average power consumption versus current.

**Table 1 entropy-24-01192-t001:** Comparison of the simulation data and experimental data [14] of pH of effluent seawater in 40 min under two currents.

Time (min)	I=30A			I=20A		
	Experimental Data [14]	Sumulated Data	Relative Error (%)	Experimental Data [14]	Sumulated Data	Relative Error (%)
5	7.92	--	--	7.95	--	--
10	7.82	7.02	10.23	7.92	7.22	8.84
15	7.07	6.58	6.93	7.63	6.79	11.01
20	6.23	6.04	3.05	6.77	6.46	4.58
25	3.27	3.37	3.06	6.28	5.99	4.62
30	2.63	2.80	6.46	3.63	3.57	1.65
35	2.46	2.57	4.47	2.93	2.98	1.71
40	2.38	2.41	1.26	2.81	2.74	2.49
Average relative error (%)			5.06			4.98

**Table 2 entropy-24-01192-t002:** $V_{\mathrm{de}}$ and $R_{\min}$ under different $i_{d}$ and $i_{0}$.

		$i_{d}({A/\mathrm{cm}}^{2})$	0.065	0.10	0.15	0.2
	$\begin{array}{l} V_{\mathrm{de}}(V)\\ /R_{\min}(\Omega ) \end{array}$					
$i_{0}({A/\mathrm{cm}}^{2})$						
10^−4^			4.8974 /0.7168	4.8708 /0.7184	4.8680 /0.7187	4.8674 /0.7188
10^−6^			4.7792 /0.7168	4.7525 /0.7184	4.7498 /0.7187	4.7491 /0.7188
10^−8^			4.6609 /0.7168	4.6343 /0.7184	4.6315 /0.7187	4.6309 /0.7188
10^−10^			4.5427 /0.7168	4.5160 /0.7184	4.5133 /0.7187	4.5126 /0.7188

**Table 3 entropy-24-01192-t003:** Variation of the equivalent inner resistances versus time.

Time (min)	$R_{i}$		
	I=30A	I=20A	I=10A
0	0.882	0.989	0.859
5	0.812	0.829	0.779
10	0.832	0.834	0.739
15	0.822	0.834	0.799
20	0.762	0.774	0.829
25	0.719	0.739	0.849
30	0.722	0.719	0.819
35	0.732	0.724	0.789
40	0.739	0.724	0.769

**Table 4 entropy-24-01192-t004:** Fit parameters under different currents.

	k	a	b	c
I=30A	0.85	30.6	−1316	$1.87\times 10^{4}$
I=20A	0.85	26.0	−1193	$1.92\times 10^{4}$

**Table 5 entropy-24-01192-t005:** Comparison of the simulated data and experimental data [14] of cell voltage of Ionpure EAC seawater in 40 min.

Time (min)	I=30A			I=20A		
	Experimental Data [14]	Simulated Data	Relative Error (%)	Experimental Data [14]	Simulated Data	Relative Error (%)
0	31.5	30.54	3.05	24.8	22.03	11.17
5	29.4	30.48	3.67	21.6	21.99	1.80
10	30.0	30.19	0.63	21.7	21.83	0.60
15	29.7	29.39	1.04	21.7	21.40	1.38
20	27.9	27.94	0.14	20.5	20.64	0.68
25	26.6	26.73	0.49	19.8	19.81	0.05
30	26.7	26.59	0.41	19.4	19.39	0.05
35	27.0	26.91	0.33	19.5	19.40	0.51
40	27.2	27.35	0.55	19.5	19.58	0.41
Average relative error (%)			1.14			1.85

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
