# Peer review of "Finite-Time Thermodynamic Modeling and Analysis of Seawater Acidification Process in Electrochemical Acidification Cell"

_entropy, 2022, doi:10.3390/e24091192_

Round 1
Reviewer 1 Report
The work by Wang et al. is a combined theory and experiment study of a seawater acidification electrochemical cell. Such a cell is of interest to study, but I had several major comments on the manuscript as follows:
- 1) There seems to be a missing component to the model. The authors consider the source of H+ in Eq. 6 from the carbonic acid reaction, but don't include the source of H+ from the water splitting reaction: H2O <--> OH- + H+ . This water splitting reaction can contribute quite significantly to H+ concentration.
- 2) Connected to point #1 above, there are several works on modeling of pH in membrane-based electrochemical system which are relevant here (can help the authors improve their model), and I believe should be cited in the manuscript as relevant prior work. A partial list:
o Atlas, I., et al. "Spatial variations of pH in electrodialysis stacks: Theory." Electrochimica Acta 413 (2022): 140151.
o Dykstra, J. E., et al. "Theory of pH changes in water desalination by capacitive deionization." Water research 119 (2017): 178-186.
Note these works cited above include the water splitting reaction in the model.
- 3) There is generally a poor presentation of data. For example, in figure 4, the theory curves do not seem smooth, which suggests they were plotted with poor resolution. The experimental data is plotted as a jagged line, but instead should be plotted with markers showing the measured data location and error bars to demonstrate repeatability.
- 4) I suspect the cell studied here will have a severe mass transport limitation due to the limited availability of H+ in the anode compartment (to carry current through the ion exchange membrane), which may be the limiting factor in the cell performance. For example, other systems with similar architecture (although different outputs) showed such limitations, which may be relevant prior literature as well:
o Abdalla S, at al. Voltage loss breakdown in desalination fuel cells. Electrochemistry Communications. 2021 Nov 1;132:107136.
o Abu Khalla, Shada, et al. "Desalination Fuel Cells with High Thermodynamic Energy Efficiency." Environmental Science & Technology 56.2 (2021): 1413-1422.

Reviewer 2 Report
The paper develops the unsteady process of the acidification of seawater by using electrochemical acidification cell. The theoretical basis of analysis is the finite time thermodynamics.
The paper is very interesting, but I must ask some improvements to the authors:
- In relation to the empirical formula: the shapes seems to be logistic. Is it possible to fit them with the logistic expression?
- The possible use of the seawater for application could be evaluated also from a thermoeconomic viewpoint: I suggest to consider the Thermodynamic Human Development Index to introduce some comments on the possible sustainable use of seawater (consider for example the papers of Grisolia, et al., recently published on Sustainability, Fronties in Energy, etc.)
After these improvements I consider that the papar could be published.
Round 2
Reviewer 1 Report
The authors have improved the manuscript by making clearer the model assumptions and deepening its discussion. Figures are also improved and citations to other models capturing pH effects in electrochemical systems were added. I have no further comments,
Reviewer 2 Report
I suggest to accept the paper.